# Isolation of Echimidine and Its C-7 Isomers from *Echium plantagineum* L. and Their Hepatotoxic Effect on Rat Hepatocytes

**DOI:** 10.3390/molecules27092869

**Published:** 2022-04-30

**Authors:** Michał Gleńsk, Marta K. Dudek, Peter Kinkade, Evelyn C. S. Santos, Vitold B. Glinski, Daneel Ferreira, Ewa Seweryn, Sławomir Kaźmierski, Joao B. Calixto, Jan A. Glinski

**Affiliations:** 1Department of Pharmacognosy and Herbal Drugs, Wroclaw Medical University, Borowska 211A, 50556 Wrocław, Poland; 2Centre of Molecular and Macromolecular Studies, Polish Academy of Sciences, Sienkiewicza 112, 90363 Łódź, Poland; mdudek@cbmm.lodz.pl (M.K.D.); kaslawek@cbmm.lodz.pl (S.K.); 3Planta Analytica LLC, 461 Danbury Rd., New Milford, CT 06776, USA; kinkadeanalytical@gmail.com (P.K.); v.glinski@plantaanalytica.com (V.B.G.); 4Centro de Inovação e Ensaios Pré-Clínicos (CIEnP), Av. Luiz Boiteux Piazza, 1302-Cachoeira do Bom Jesus, Florianópolis 88056000, SC, Brazil; evelyn.santos@cienp.org.br (E.C.S.S.); joao.calixto@cienp.org.br (J.B.C.); 5Department of BioMolecular Sciences, Division of Pharmacognosy, Research Institute of Pharmaceutical Sciences, School of Pharmacy, University of Mississippi, Oxford, MI 38677, USA; dferreir@olemiss.edu; 6Department of Chemistry and Immunochemistry, Wroclaw Medical University, M. Skłodowskiej-Curie 48/50, 50369 Wrocław, Poland; ewa.seweryn@umw.edu.pl

**Keywords:** echimidine, echimidine isomers, hepatotoxicity, *Echium plantagineum* L., rat hepatocytes primary culture

## Abstract

Echimidine is the main pyrrolizidine alkaloid of *Echium plantagineum* L., a plant domesticated in many countries. Because of echimidine’s toxicity, this alkaloid has become a target of the European Food Safety Authority regulations, especially in regard to honey contamination. In this study, we determined by NMR spectroscopy that the main HPLC peak purified from zinc reduced plant extract with an MS [M + H]^+^ signal at *m*/*z* 398 corresponding to echimidine (**1**), and in fact also represents an isomeric echihumiline (**2**). A third isomer present in the smallest amount and barely resolved by HPLC from co-eluting (**1**) and (**2**) was identified as hydroxymyoscorpine (**3**). Before the zinc reduction, alkaloids (**1**) and (**2**) were present mostly (90%) in the form of an *N*-oxide, which formed a single peak in HPLC. This is the first report of finding echihumiline and hydroxymyoscorpine in *E. plantagineum*. Retroanalysis of our samples of *E*. *plantagineum* collected in New Zealand, Argentina and the USA confirmed similar co-occurrence of the three isomeric alkaloids. In rat hepatocyte primary culture cells, the alkaloids at 3 to 300 µg/mL caused concentration-dependent inhibition of hepatocyte viability with mean IC_50_ values ranging from 9.26 to 14.14 µg/mL. Our discovery revealed that under standard HPLC acidic conditions, echimidine co-elutes with its isomers, echihumiline and to a lesser degree with hydroxymyoscorpine, obscuring real alkaloidal composition, which may have implications for human toxicity.

## 1. Introduction

Pyrrolizidine alkaloids (PAs) are toxic compounds widespread throughout the plant kingdom, occurring in about 3% of flowering plants [1]. Over 350 PAs have been identified so far [2]. The plants containing PAs belong mainly to the Asteraceae (Senecioneae and Eupatorieae tribes), Boraginaceae (all genera), and Fabaceae (genus Crotalaria) families [3,4,5,6]. PAs occur as free-base/tertiary forms or their *N*-oxides. Both forms are hepatotoxic and genotoxic [7,8]. Often, the *N*-oxides are found in higher concentrations than the corresponding free bases (tertiary PAs) [9,10].

The well-documented liver toxicity of 1,2-unsaturated PAs results in hepatic veno-occlusive disease (HVOD). Although chronic exposure to PAs may not produce readily recognizable symptoms, their hepatotoxicity can lead to irreversible liver damage and death [5,6].

Because PAs may enter into the food chain by consumption of teas and herbal products, eggs, milk, cereals and honey, awareness was recently raised by the European Food Safety Authority (EFSA) [5,6,11,12,13]. Although legal limits for PAs in food have not been yet established in the European Union, the German Federal Institute for Risk Assessment (BfR) recommended a limit for intake of no higher than 0.007 μg of 1,2-unsaturated PAs per day per kg body weight [14].

Certain PAs-containing plants such as *Echium plantagineum* and *E. vulgare* are important foraging sources for honeybees. The Food Standards Australia New Zealand (FSANZ) recognizes that honey produced from *E. plantagineum* will contain PAs and suggests blending it with other kinds of honey to reduce PAs concentration or that consumers should not exclusively eat this honey [10,13].

The main pyrrolizidine alkaloid of *E. plantagineum* L., known as Paterson’s curse or Salvation Jane in Australia, is echimidine [15,16,17]. This plant is native to the Mediterranean basin in Spain and Portugal and is naturalized across much of southern Australia. It was originally introduced as a potherb in the mid-1800s and later spread as an unintended contaminant of pasture seed and hay. *E. plantagineum* is estimated to have invaded over 30 million ha of grazing land in Australia [18,19]. In addition to the fact that *E. plantagineum* contains pyrrolizidine alkaloids, its traditional uses as diaphoretic diuretic, cough and wound healing agent is mentioned. Moreover, echium oil also has many potential uses in cosmetic and personal care product industries [20].

The presence of PAs in honey is well documented [10,13], but at what concentration level could they be considered dangerous is still being debated. Currently, there are several research programs underway, initiated by countries that either export or import the honey, aimed at the determination of toxic limits for pyrrolizidine alkaloids. For most of the studies, echimidine has been chosen as one of the representative alkaloids.

In this paper, we describe the isolation and determination of three isomeric pyrrolizidine alkaloids from *E. plantagineum,* including echimidine, echihumiline and hydroxymyoscorpine, the latter two being isolated from this source for the first time. The latter compounds have been previously isolated from other plants [4,17]. Considering that these alkaloids may contribute unequally to the hepatotoxicity effects, the “echimidine” peak was further resolved into true echimidine (**1**) and the mixture of echihumiline (**2**) and hydroxymyoscorpine (**3**) and their cytotoxicity were assessed in rat primary cell hepatocytes. Additionally, the hepatotoxic activity of echiumine (**4**), the alkaloid isolated from the same plant material as the three aforementioned isomeric alkaloids, was also evaluated.

## 2. Results and Discussion

### 2.1. Chromatography Investigation

Echimidine is currently used as a standard for the determination of toxic limits for pyrrolizidine alkaloids in many products; therefore, its purity is of great importance. In our study, we noticed that the methanol extract contained nearly 90% of echimidine, as the *N*-oxide represented by a sharp RP-HPLC peak. The *N*-oxides were reduced with zinc dust in the presence of sulfuric acid. After the reduction, the alkaloidal fraction was collected by extraction with ethyl acetate. The alkaloid mixture was subjected to a CPC purification step that produced a fraction composed of echimidine plus its isomers. An RP HPLC analysis of this fraction under acidic conditions (0.05% TFA) consistently produced a sharp peak plus a small 1–2% area partially resolved peak eluting ahead of the main peak showing an MS ion at *m*/*z* 398 [M + H]^+^ corresponding to echimidine (Figure 1). Considering the HPLC profile and keeping the MS result in mind, the sample may be regarded as an echimidine of good purity.

However, further NMR and HPLC data analysis indicated that this single peak comprises a mixture of alkaloids. Using the “core-shell” RP HPLC column (Kinetex EVO C18, Phenomenex) in a buffer system at pH 6.8 (or higher), the peak attributed to “echimidine” was resolved into two well-separated peaks (Figure 2). Peak A consists of echihumiline and hydroxymyoscorpine, whereas peak B represents echimidine.

Additionally, with the aim of centrifugal partition chromatography (CPC) in a biphasic system, consisting of chloroform as the mobile phase and citrate buffer at pH 5.6 as the stationary phase, we were able to simultaneously collect a fraction that contained mainly echiumine (**4**) (Figure 3), an alkaloid closely related to echimidine.

### 2.2. NMR Analysis

The NMR data analysis proved that peak **B** is indeed echimidine, while peak **A** comprised of two largely unresolved alkaloids, echihumiline (major) and hydroxymyoscorpine (minor). Examination of several older (highly enriched in echimidine) comparable fractions derived from different *E. planatgineum* plant collections revealed that the ratio of echihumiline to echimidine varied from 0.13 to 0.43. The ratio of hydroxymyoscropine to echimidine varied from non-detectable to 0.02. Nevertheless, the earlier eluting peak **A** in Figure 2, which in fact is a 3:1 mixture consisting of two alkaloids by NMR data analysis, could not be fully resolved through pH-controlled systems or the use of various stationary phases (C_8_, C_12_, C_18_).

A detailed analysis of the ^1^H NMR spectrum of the constituent comprising peak **A** clearly indicates that two compounds, in the ratio of 3:1, are present. This was evaluated on the basis of integral values of the well-separated ^1^H resonances, which are directly proportional to the number of protons giving rise to a given signal. The spectroscopic pattern of both compounds is characteristic of pyrrolizidine alkaloids, with many signals resonating at the same frequencies. However, broad singlets at δ_H_ 4.48 and 4.44, assignable to H-8 of the major and minor components (Table 1), respectively, are well separated. There is also a difference in the ^1^H and ^13^C chemical shifts of the C-18-C-21 moieties. In the minor component, the ^1^H signals at δ_H_ 6.75, 1.75 and 1.76, together with their adjacent carbons, are characteristic of a tigloyl moiety and assignable to C-19, C-20 and C-21 sites (Table 1), respectively, implying that this component is hydroxymyoscorpine (Figure 3) [21]. In the ^1^H NMR spectrum of the major component, the olefinic proton (H-18) signal is shielded to δ_H_ 5.57, in contrast to H-19 of echimidine and hydroxymyoscorpine, in which these signals resonate at 6.75 and 6.09, respectively. In addition, both methyl groups are deshielded to δ_H_ 1.88 and 2.12, in comparison to 1.75 and 1.76 ppm seen for hydroxymyoscorpine (see Table 1). These chemical shifts are characteristic of a senecioic acid ester moiety, which permits the identification of the major component represented by peak **A** as echihumiline (Figure 3) [22].

The ^1^H NMR spectrum of the second compound comprising peak **B** exhibits five methyl, four methylene, two olefinic and three methine proton signals. The chemical shifts at δ_H_ 6.09 and 5.85 (each 1^-^H, corresponding to H-19 and H-2, see Table 1) are characteristic of olefinic moieties. One of these signals (δ_H_ 6.09) correlates in the HMBC spectrum with two methyl signals at δ_C_ 20.48 and 15.78 (C-21 and C-20, respectively), and one carboxyl signal at δ_C_ 166.75 (C-17). The protons adjacent to the two methyl groups resonate at δ_H_ 1.79 (H-21) and 1.94 (H-20), respectively, and in conjunction with a carboxylic carbon at δ_C_ 166.75 (C-17), an sp^2^ quaternary carbon signal at δ_C_ 127.21 (C-18), and a methine carbon signal at δ_C_ 139.63 (C-19) are all characteristic of an angeloyl moiety [23]. The location of this moiety is indicated by an HMBC correlation of the carboxylic carbon with a signal at δ_H_ 5.45 (H-7, Table 1) belonging to the pyrrolizidine moiety, as indicated by additional HMBC and COSY correlations. A second side-chain moiety is substituted with a carbonyl group resonating at δ_C_ 174.24 and exhibiting long-range correlation to a proton signal at δ_H_ 4.17. This signal shows further correlations to a methyl group at δ_C_ 18.48 and two oxygenated tertiary carbons at δ_C_ 73.73 and 83.07. Both these carbons give HMBC cross-peaks to two methyl signals at δ_C_ 24.77 and 25.98. As a result, the compound representing peak B was unambiguously identified as echimidine (Figure 3). All ^1^H and ^13^C chemical shifts for echimidine and its isomers are collated in Table 1 and are in perfect agreement with published data [4,17,24,25].

The apparent similarity of the ^1^H and ^13^C NMR spectra of these alkaloids (see Figure 4) may be a reason why their existence has not been noticed before in what was regarded previously as a pure “echimidine” peak. The presence of a nitrogen atom, which is easily protonated at lower pH, and as a result may cause significant changes to proton chemical shifts, may further interfere with the proper identification, especially when the spectra are often registered in CDCl_3_, which may have a different degree of acidity.

### 2.3. Hepatotoxicity Assessment

Figure 5A shows the percentage of viability of hepatocytes treated with different concentrations of echiumine, a fraction with an “echimidine” peak (being in fact a mixture of echimidine, echihumiline and hydroxymyoscorpine), echimidine and a mixture of echihumiline and hydroxymyoscorpine (3 to 300 µg/mL).

All of them caused a concentration-dependent inhibition of rat hepatocyte primary cell culture viability. The calculated mean IC_50_ values for the four compounds ranged from 7.47 to 14.14 µg/mL and they do not differ significantly among them (Table 2). As a positive control (Figure 5B), we assessed the effect of two hepatotoxic agents, namely acetaminophen and ethanol at the concentrations of 15.1 and 80 µL/mL, respectively. Both agents inhibited hepatocyte viability in 70.68% and 72.51%, respectively, confirming that the bioassay worked properly.

Collectively, the results support the view that the alkaloids isolated from *Echium plantagineum* L. exhibit similar hepatotoxicity when assessed in rat hepatocyte primary culture cells. Importantly, the IC_50_ value of isolated echimidine (13.79 µg/mL) was similar to that of the fraction containing the three isomers (14.14 µg/mL). However, the highest toxicity was found for echiumine (7.47 µg/mL), which was isolated alongside with the three aforementioned alkaloids from the same plant sample. According to some authors, the pyrrolizidine alkaloids are converted in the liver into reactive metabolites. Therefore, chronic exposure to sub-lethal doses may cause cumulative damage or cancer. PAs intake can induce damage to liver cells, inducing hepato-sinusoidal obstruction syndrome or veno-occlusive disease. Testing of a range of PAs revealed that nuclear receptor PXR was exclusively activated by the open-chain diesters such as echimidine and lasiocarpine, suggesting that only open-chain diesters act as PXR agonists. This might imply that a PXR-dedicated mode of action may contribute to the hepatotoxicity of PAs that is dependent on PA structure [26].

## 3. Materials and Methods

### 3.1. General Experimental Procedures

Centrifugal Partition Chromatography was performed on the FCPC Kromaton A100 (Rousellet Robatel, Annonay, France) system. Analytical HPLC was carried out on an Agilent HP1100 model equipped with quaternary pump, DAD, and autosampler. Preparative HPLC was performed on a Gilson model 333/334 with H3M pump heads. A linear UV-VIS 200 detector with a preparative cell was used. The UV signal was monitored and integrated with a model N2000 Baseline Chromatography Data System from Baseline Chromatech Research Centre (www.qinhuan.com, accessed on 24 March 2022). Analytical HPLC analyses were performed on a Kinetex EVO C_18_, 5 µm, 100 Å, 4.6 × 150 mm column, and preparative on a Kinetex EVO C_18_, 5 µm, 100 Å, 21.2 × 150 mm column (Phenomenex, Torrance, CA, USA). The mobile phase buffer was the same as for the preparative HPLC (32 mM lithium phosphate, adjusted to pH 7.2 with phosphoric acid). The samples/compounds were eluted from the analytical column on a gradient from 5 to 16% acetonitrile in 15 min, followed by a column wash with 29% acetonitrile. The flow rate was 1.0 mL/min. The UV absorbance signal on the diode array detector was monitored at 210 nm.

### 3.2. Plant Material

The plant material was collected in New Zealand, and the voucher specimens (CHR 637254, CHR 637255) were deposited at Allan Herbarium (CHR), Landcare Research, PO Box 69040, Lincoln 7640, NZ. The American Echium was propagated from the seeds sold by Outsidepride company, 915 N. Main St., Independence, OR 97351, in Bethel, Connecticut. Argentinian *E. plantagineum* was collected in B6600 Mercedes, Provincia de Buenos Aires, Argentina.

### 3.3. Extraction and Isolation

Collected in New Zealand, aerial parts of *E. plantagineum* were dried and then extracted with MeOH. After the removal of MeOH, the resulting oleoresin was defatted by partitioning between *n*-hexane-MeOH-water (10:9:1). The lower phase was concentrated to remove the organic solvents and diluted with water. The *N*-oxides of the Pas were reduced to free alkaloids by treating the aqueous phase with 50% H_2_SO_4_ till the pH adjusted to 2, and then an excess of zinc dust was added during stirring. In about 6 h the pH became nearly neutral, and another portion of sulfuric acid and zinc dust was added. After 20 h of stirring, the reaction mixture was made alkaline with Na_2_CO_3_, and the alkaloids were extracted several times with ethyl acetate containing 5% of isopropanol. These extracts were combined and quickly evaporated to dryness to avoid hydrolysis of ethyl acetate by the alkaloids. Next, the obtained solid was again partitioned between water acidified with sulfuric acid and ethyl acetate. The alkaloids migrated into the aqueous acidic phase, and the neutral compounds remained in the ethyl acetate phase. Finally, the aqueous layer was made alkaline with Na_2_CO_3_ and the alkaloids were re-extracted into ethyl acetate. Evaporation of the solvent produced a crude mixture of pyrrolizidine alkaloids. Centrifugal Partition Chromatography, using chloroform as the mobile phase and citrate buffer at pH 5.6 as the stationary phase, was used to produce the bands of echimidine fraction and echiumine. After this step, the echimidine fraction was purified by preparative HPLC. The mobile phase contained a buffer of 32 mM lithium phosphate, adjusted to pH 7.2 with phosphoric acid. The column was equilibrated using a 9:1 mixture of the buffered mobile phase and acetonitrile and run in a gradient from 10% to 20% MeCN over 14.4 min at 20 mL/min. For preparative runs, samples of 60 mg were injected in methanol/water/acetic acid (1:1:0.005) at a concentration of 100 mg/mL. A total of 1.3 g of CPC-fractionated echimidine fraction was loaded in 22 preparative HPLC runs. As a result, 630 mg of echimidine and 280 mg of echihumiline/hydroxymyoscorpine mixture were obtained.

### 3.4. NMR Spectroscopy

The 1D and 2D NMR spectra were recorded on a Bruker Avance III 500 spectrometer (Bruker BioSpin, Rheinstetten, Germany), operating at 500.13 MHz for ^1^H and 125.76 MHz for ^13^C. The spectrometer was equipped with a 5 mm BBI probe head with an actively shielded Z-gradient coil connected to a GAB/2 gradient unit capable of producing B_0_ gradients with a maximum strength of 50 G/cm. During all measurements, the temperature was set at 295 K and was controlled and stabilized with a BCU 05 cooling unit controlled by a BVT3200 variable temperature unit. All spectra were recorded using 3 mm NMR tubes (Norell) and CDCl_3_ (Armar Chemical, Döttingen, Switzerland) as a solvent. Chemical shifts were referenced to the residual solvent signals at 7.26 and 77.00 ppm for ^1^H and ^13^C, respectively. All spectra, except long-range ^1^H-^13^C correlation spectra, were acquired with the original Bruker pulse sequences. For observation of the long-range correlations, the Impact-HMBC pulse sequence was used. The spectra were acquired and processed using the TopSpin 3.1 program (Bruker BioSpin) running under Windows 7 (64 bit) OS on the HP Z700 workstation and used for operating and controlling the spectrometer.

### 3.5. Hepatotoxicity Assay

#### 3.5.1. Animals

Male Sprague Dawley (250–350 g) rats were used throughout this study. Animals originated from Charles River Laboratories (Raleigh, NC, USA) were bred in the Centre for Innovation and Pre-Clinical Studies animal house facility (Florianópolis-S.C., Brazil). All animals were Specific Pathogen Free (5 per cage) and were housed at 22 ± 1 °C in a light-controlled environment under a 12–12-h light-dark cycle (lights on at 07:00 AM). All experimental procedures were based on the “Principles of Laboratory Animal Care” from the NIH publication No. 85-23 [27], the international standards of animals recommended by Brazilian Law (# 11.794–10/08/2008) [28] and were approved by the Animal Ethics Committee of the CIEnP—Centre of Innovation and Pre-Clinical Studies (#243/00/CEUA/CIEnP).

#### 3.5.2. Hepatocyte Isolation and Cultivation

Hepatocytes were isolated from male Sprague-Dawley rats by a modified two-step collagenase perfusion method [29]. The viability of hepatocytes was >80% (determined by trypan blue exclusion). Hepatocytes were cultured in a 48 well plate at a density of 2 × 10^5^ cells/well in William’s medium supplemented with 5% fetal bovine serum and 1% penicillin/streptomycin under standard conditions in a humidified atmosphere at 37 °C and 5% CO_2_ for 4 h. The medium was removed and replaced by HepatoZyme. After 18 h of plating, cells were treated with the compounds echiumine, echimidine fraction, echimidine (isolated) or the echihumiline/hydroxymyoscorpine mixture at concentrations of 3, 10, 30, 100 and 300 µg/mL. Acetaminophen (15.1 mg/mL) and ethanol (63.2 µg/mL) were used as positive controls. The effects of the compounds were evaluated for 48 h.

#### 3.5.3. MTT Assay

MTT assay viability was performed 48 h after the treatment of cells and was assessed by measuring the formation of formazan from the MTT spectrophotometric test, according to Mosmann (1983) [30]. At the end of the experiment, the hepatocytes were incubated with 0.5 mg/mL MTT for 4 h min at 37 °C. After the MTT was removed and the blue formazan was extracted from cells with DMSO (100 µL/well). The absorbance was read at 570 nm in a microplate reader (SpectraMax Plus).

#### 3.5.4. Data Analysis

Graphic data were expressed as mean ± SEM. Statistical evaluation of the results was carried out using the appropriate one-way analysis of variance (ANOVA) followed by a post-hoc Dunnett’s test, whereas control groups (ACP or EtOH) were analyzed by Student’s *t*-test. *p* values lower than 0.05 (* *p* < 0.05) were considered statistically significant. Data were analyzed using GraphPad^®^ Prism 5.0 software (San Diego, CA, USA).

## 4. Conclusions

For the first time, we have proven that the fraction with an “echimidine” peak in RP HPLC derived from *E. plantagineum* consists of three alkaloids: echimidine, echihumiline, and hydroxymyoscorpine. We believe that the two unreported alkaloids, echihumiline and hydroxymyoscorpine, have been overlooked in this species because MS does not differentiate isomers. Furthermore, as a result of significant structural similarity, most of the ^1^H NMR signals are overlapping, and thus, signals of echihumiline and hydroxymyoscorpine, which are at lower concentrations, can be easily overlooked. We surmise that their presence varies and may be limited to some varieties, or they may occur at low concentrations compared to echimidine. Additionally, the present results support the view that the alkaloids isolated from *Echium plantagineum* L. exhibit similar hepatotoxicity when assessed in rat hepatocyte primary culture cells.

## Figures and Tables

**Figure 1 molecules-27-02869-f001:**
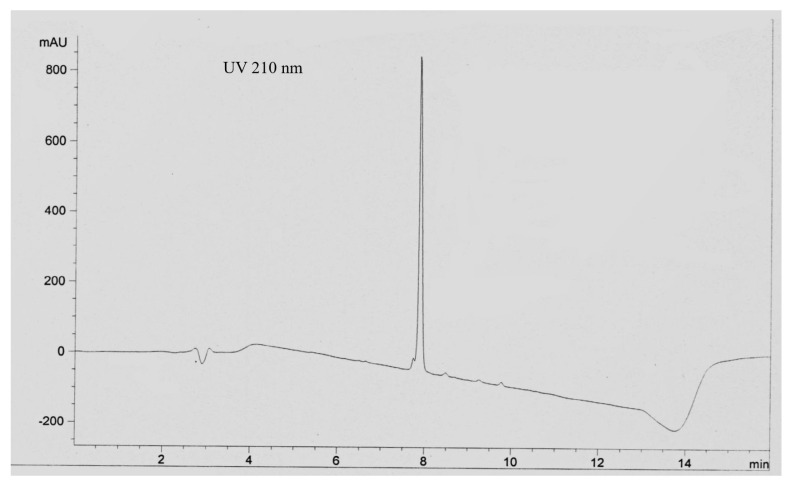
An RP HPLC analysis of the “echimidine” peak from *E. plantagineum* under acidic conditions (0.05% TFA). This peak in fact consists of Echimidine, Echihumiline and Hydroxymyoscorpine.

**Figure 2 molecules-27-02869-f002:**
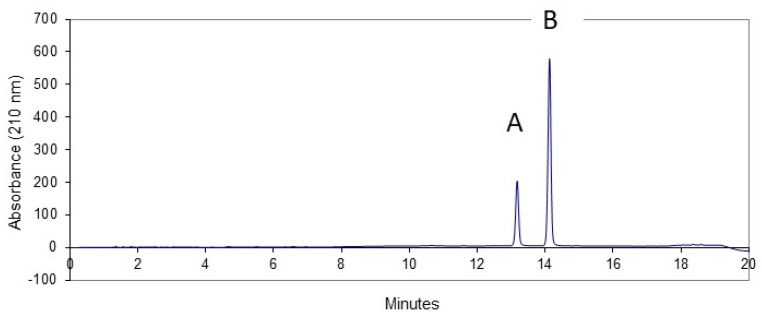
An RP HPLC analysis of the “echimidine” peak from *E. plantagineum* under buffer conditions (pH 6.8). Peak A—mixture of echihumiline and hydroxymyoscorpine. Peak B—echimidine.

**Figure 3 molecules-27-02869-f003:**
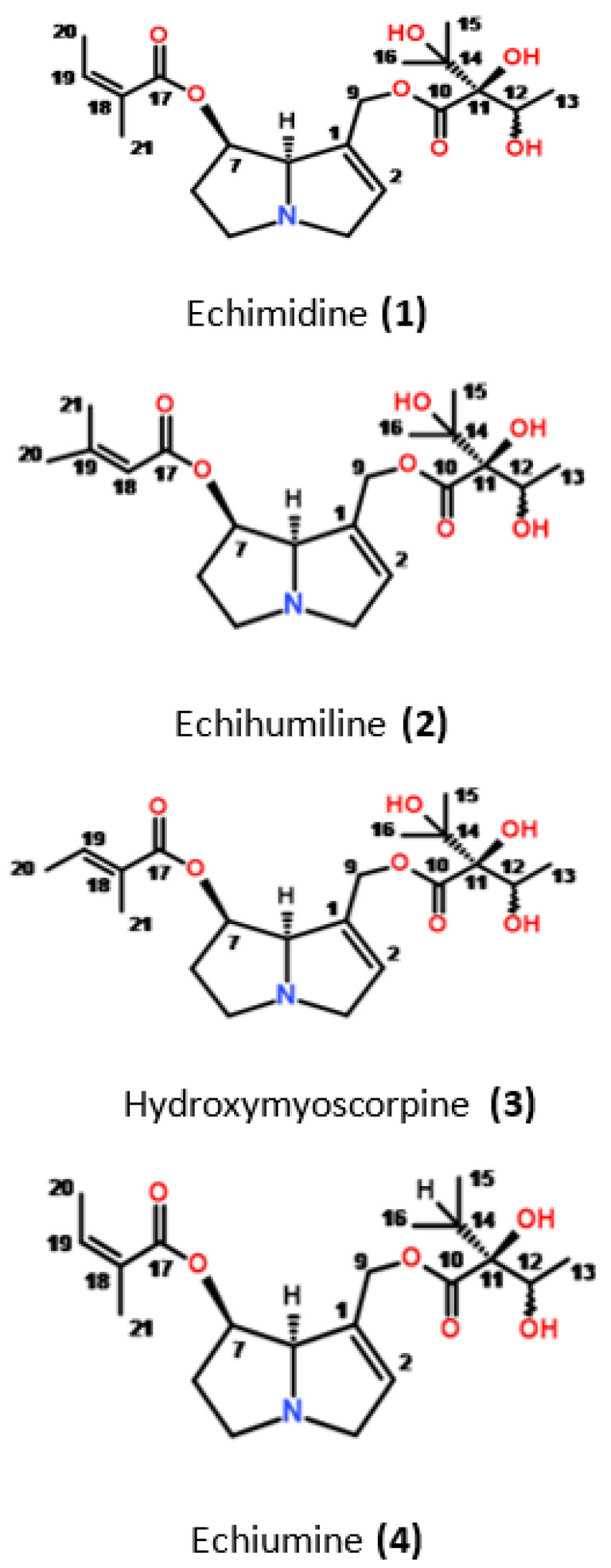
Structures of the investigated alkaloids (Stereochemistry at position C-12 not established).

**Figure 4 molecules-27-02869-f004:**
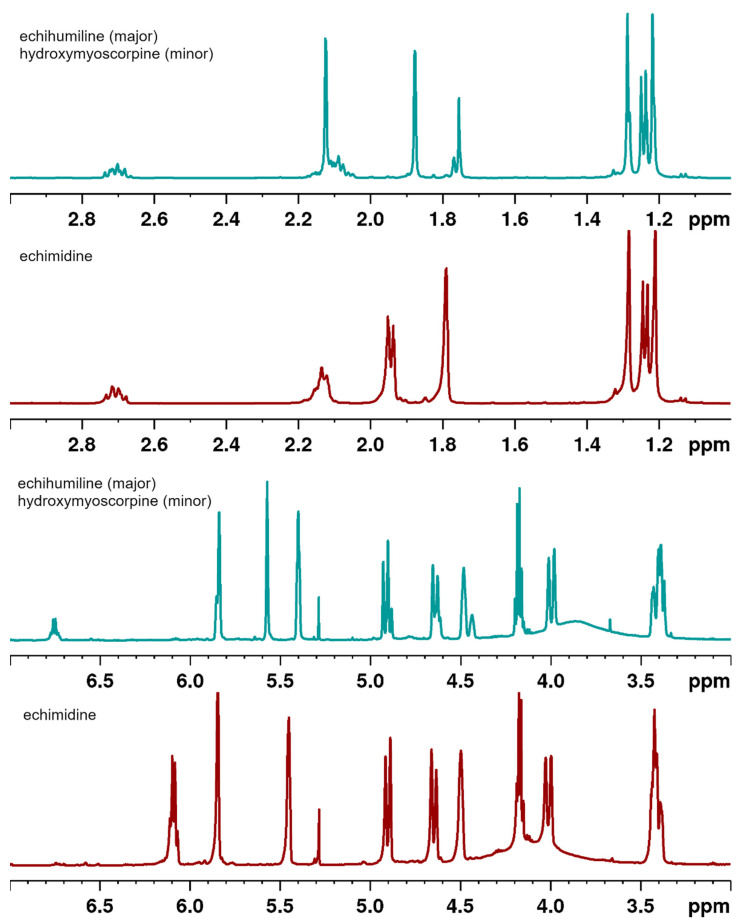
^1^H NMR spectra of echimidine and echihumiline-hydroxymyoscorpine mixture (CDCl_3_, 500 MHz).

**Figure 5 molecules-27-02869-f005:**
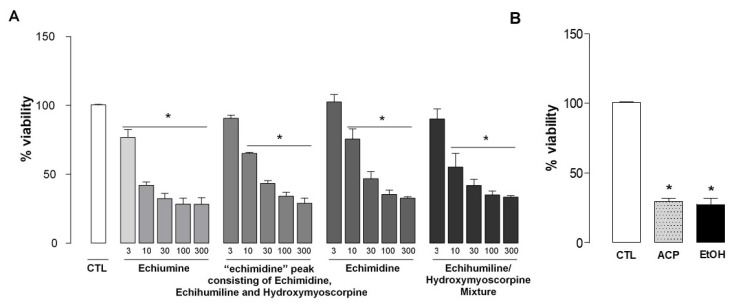
Effect of the compounds on cell viability in primary culture of rat hepatocytes. (**A**) Rat hepatocytes were treated with echiumine, a fraction with an “echimidine” peak, echimidine or echihumiline/hydroxymyoscorpine mixture (3–300 µg/mL) for 48 h. (**B**) Rat hepatocytes were treated with acetaminophen (ACP; 15.1 mg/mL) or ethanol (EtOH; 80 µL/mL) as a positive control. Cell viability was determined by MTT assay. The results are expressed in percentage of control (CTL) and each column presents the mean and the SEM of triplicate assays of two independent experiments. * *p* <0.05 when compared with control (CTL) values (one-way ANOVA or *t*-test, when applicable).

**Table 1 molecules-27-02869-t001:** NMR data for echihumiline, hydroxymyoscorpine and echimidine (CDCl_3_, 500 MHz, T = 295 K).

Position	Peak (A)				Peak (B)	
	Major Component (1.00) Echihumiline		Minor Component (0.33) Hydroxymyoscorpine		Echimidine	
	δ (^13^C), Multiplicity	δ (^1^H) *J* in Hz	δ (^13^C), Multiplicity	δ (^1^H) *J* in Hz	δ (^13^C), Multiplicity	δ (^1^H) *J* in Hz
1	132.9, C		132.8, C		132.9, C	
2	128.1, CH	5.84 bs	128.2, CH	5.86 bs	128.2, CH	5.85 bs
3	62.4, CH_2_	4.00 bd (15.2)	62.7, CH_2_	4.00 bd (15.2)	62.5, CH_2_	4.01 bd (15.2)
		3.42 m		3.41 m		3.40 m
5	53.7, CH_2_	3.39 m	53.8, CH_2_	3.39 m	53.8, CH_2_	3.42 m
		2.70 m		2.70 m		2.71 m
6	34.3, CH_2_	2.11 m (2H)	34.3, CH_2_	2.11 m (2H)	34.4, CH_2_	2.13 m (2H)
7	73.6, CH	5.40 m	73.6, CH	5.40 m	73.6, CH	5.45 m
8	75.8, CH	4.48 bs	75.9, CH	4.44 bs	75.9, CH	4.50 bs
9	62.3, CH_2_	4.92 d (13.1)	62.5, CH_2_	4.89 (overlapped)	62.3, CH_2_	4.90 d (13.2)
		4.64 d (13.1)		4.63 d (13.2)		4.65 d (13.2)
10	174.2, C		174.2, C		174.2, C	
11	82.9, C		82.9, C		83.1, C	
12	69.7, CH	4.18 q (6.3)	69.7, CH	4.18 q (6.4)	69.7, CH	4.17 q (6.4)
13	18.5, CH_3_	1.24 d (6.3)	18.5, CH_3_	1.24 d (6.4)	18.5, CH_3_	1.24 d (6.4)
14	72.7, C		72.7, C		73.7, C	
15	25.9, CH_3_	1.22 s	25.9, CH_3_	1.21 s	26.0, CH_3_	1.21 s
16	24.8, CH_3_	1.29 s	24.8, CH_3_	1.28 s	24.8, CH_3_	1.28 s
17	165.7, C		167.0, C		166.8, C	
18	115.5, CH	5.57 m	128.4, C		127.2, C	
19	158.4, C		138.0, CH	6.75 qd (6.2; 1.5)	139.6, CH	6.09 qq (7.3; 1.6)
20	27.5, CH_3_	1.88 d (0.9)	14.5, CH_3_	1.76 dd	15.8, CH_3_	1.94 dd (7.3; 1.6)
21	20.3, CH_3_	2.12 d (0.9)	11.9, CH_3_	1.75 bs	20.5, CH_3_	1.79 d (1.6)

**Table 2 molecules-27-02869-t002:** Mean IC_50_ and maximal inhibitions values on cell viability in primary culture hepatocytes for different compounds.

Compound	IC_50_	95% CI	I_max_ (%)
“echimidine” peak consisitng of Echimidine, Echihumiline and Hydroxymyoscorpine	14.14 µg/mL	9.01 to 22.17	70.96
Echimidine	13.79 µg/mL	7.84 to 24.24	67.37
Echihumiline/Hydroxymyoscorpine Mixture	9.26 µg/mL	4.33 to 19.81	66.51
Echiumine	7.47 µg/mL	3.26 to 17.11	71.64
Acetaminophen	3.82 mg/mL	2.0 to 7.32	68.31

IC_50_ = concentration required to inhibit the cytotoxicity by 50%; CI = the 95% confidence interval, I_max_ = maximum inhibition of cell viability. Each group represents the triplicate of two independent animals.

## Data Availability

Data sharing not applicable.

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
