# Peer review of "Isolation of Echimidine and Its C-7 Isomers from Echium plantagineum L. and Their Hepatotoxic Effect on Rat Hepatocytes"

_molecules, 2022, doi:10.3390/molecules27092869_

Round 1
Reviewer 1 Report
The manuscript titled “Isolation of Echimidine and its C-7 isomers from Echium plantagineum L. and their hepatotoxic effect on rat hepatocytes” brings a significant contribution as it identifies toxic isomeric alkaloids that had been overlooked before in this particular plant material. The manuscript is well-written, the methodology is adequate, and the language is appropriate. I only have a couple of comments that should be addressed in order to enhance the paper even further. Therefore, I recommend minor revisions.
Abstract: the abstract is missing a closing sentence highlighting the importance of identifying these alkaloids.
Comment: More insight into the toxicity mechanisms of the identified alkaloids should be given in the discussion.
Author Response
Comments and Suggestions for Authors (1)
The manuscript titled “Isolation of Echimidine and its C-7 isomers from Echium plantagineum L. and their hepatotoxic effect on rat hepatocytes” brings a significant contribution as it identifies toxic isomeric alkaloids that had been overlooked before in this particular plant material. The manuscript is well-written, the methodology is adequate, and the language is appropriate. I only have a couple of comments that should be addressed in order to enhance the paper even further. Therefore, I recommend minor revisions.
Abstract: the abstract is missing a closing sentence highlighting the importance of identifying these alkaloids.
We did add the following sentence to the abstract:
Our discovery revealed that under standard HPLC acidic conditions echimidine co-elutes with its isomers, echihumiline and to a lesser degree with hydroxymyoscorpine, obscuring real alkaloidal composition, which may have implications for human toxicity.
Comment: More insight into the toxicity mechanisms of the identified alkaloids should be given in the discussion.
We did add the following sentence to the discussion:
According to some authors the pyrrolizidine alkaloids are converted in the liver into reactive metabolites. Therefore chronic exposure to sub-lethal doses may cause cumulative damage or cancer. PAs intake can induce damage to liver cells, inducing hepato-sinusoidal obstruction syndrome or veno-occlusive disease. Testing of a range of PAs revealed that nuclear receptor PXR was exclusively activated by the open-chain diesters such as echimidine and lasiocarpine, suggesting that only open-chain diesters act as PXR agonists. This might imply that a PXR-dedicated mode of action may contribute to the hepatotoxicity of PAs that is dependent on PA structure [26].
and the reference:
[26] Luckert, C.; Braeuning, A.; Lampen, A.; Hessel-Pras, S. PXR: Structure-specific activation by hepatotoxic pyrrolizidine alkaloids. Chem. Biol. Interac. 2018, 288, 38-48.
Reviewer 2 Report
The relevance of the manuscript “Isolation of Echimidine and its C-7 isomers from Echium plantagineum L. and their hepatotoxic effect on rat hepatocytes” is that it calls for the importance to improve analytical methods to detect the pyrrolidizidine alkaloids from Echium species. Nevertheless, to highlight the potential of the reported discoveries, the expression of the results has to be improved.
Line 34- “Retroanalysis of our samples of E. plantagineum collected in New Zealand, Argentina and the USA confirmed similar co-occurrence of the three isomeric alkaloids.” The authors do not present data concerning the concentration of the 4 compounds on the different samples (only present ratios) and there is no reference to the origin of theses samples on the Materials and methods section.
Line 99- “peak showing an MS ion at m/z 398”. Please indicate the ionization mode. Does it correspond to a protonated molecule or refers to other mass adduct?
Line 105- The quality of figure 1 needs to be improved:
- Remove the colored background
- Remove the text box from the figure
- Add the units to the axis
- Improve the contents of the figure´s legend
Line 115- The quality of figure 2 needs to be improved:
- Remove the colored background
- Remove the text and the arrows, leave only the letter attributed to each peak “A” or “B”
- Remove the outer lines
- Improve the contents of the figure´s legend
Line 120- “An NMR data analysis”, maybe replace by The NMR data analysis
Line 129-158- The quality of the NMR description should be improved. The authors should always relate the carbon and proton chemical shifts to the atom number, also the name of the compounds should be presented as e.g. echimidine (1); echihumiline (2). Rephrase to achieve a better and less confusing description.
- “A detailed analysis of the 1H NMR spectrum of the constituent comprising peak A clearly indicates that two compounds, in the ratio of 3:1, are present.” How did you observed/calculated this?
- “In the minor component, the 1H signals at δH 6.75, 1.75 and 1.76, together with their adjacent carbons are characteristic for a tigloyl moiety, implying that this component is hydroxymyoscorpine (Figure 3)” Please assign the position to those proton chemical shifts and give a reference that corroborates your observations.
- “In the 1H NMR spectrum of the major component, the olefinic proton (H-18) signal is shielded to δH 5.57, as compared to H-19 of echimidine and hydroxymyoscorpine, whereas the methyl groups are both deshielded to δH 1.88 and 2.12. These chemical shifts are characteristic for a senecioic acid ester moiety, which permits the identification of the major component represented by peak A as echihumiline” – Improve to a clearer sentence, and provide a reference that corroborates your observations.
- “The 1H NMR spectrum of the second compound comprising peak B (…), as indicated by additional HMBC and COSY correlations”. Please assign the position of those chemical shifts and give a reference that corroborates your observations
- Make use of Table 1 throughout the text.
Line 166-- The quality of figure 3 needs to be improved:
- Increase the pixel resolution of the structures
- Add the name of the compound below each structure e.g. echimidine (1); echihumiline (2)…
- Improve the contents of the figure´s legend
Line 180 – Improve the formatting of Table 1:
- Indicate the solvent, and frequency used for the NMR experiments
- Include the word “position” on the column of 1,2,3,…
- Present the carbon NMR data with only one decimal and introduce the number of attached protons using the C, CH, CH2, and CH3 notation.
- Use a space between the proton chemical shift data and the multiplicity of the signal
- Please indicate all the coupling constants
Line 193 – Figure 4:
- Indicate the solvent, and frequency used for the NMR experiments
- Take care that the text does not overlap the peaks
Lines 195-198- Is not clear what was the chromatographic relation between compounds 1-3 and compound 4. Please include experimental data and a reference that corroborates the assignment of compound 4 as echiumine. What is the importance of this finding?
Line 247- Replace UVIS by UV-VIS
Line 272- “These extracts were combined and quickly evaporated to dryness to avoid hydrolysis of ethyl acetate by the alkaloids.” Is this correct?
Line 343 – “We believe that the two unreported alkaloids echihumiline and hydroxymyoscorpine have been overlooked in this species because MS does not differentiate isomers.” Could it be that these compounds have been overlooked because the standard experimental conditions are not appropriate like the liquid chromatography conditions used, or the lack of MS/MS data?
Author Response
Comments and Suggestions for Authors (2)
The relevance of the manuscript “Isolation of Echimidine and its C-7 isomers from Echium plantagineum L. and their hepatotoxic effect on rat hepatocytes” is that it calls for the importance to improve analytical methods to detect the pyrrolidizidine alkaloids from Echium species. Nevertheless, to highlight the potential of the reported discoveries, the expression of the results has to be improved.
Line 34- “Retroanalysis of our samples of E. plantagineum collected in New Zealand, Argentina and the USA confirmed similar co-occurrence of the three isomeric alkaloids.” The authors do not present data concerning the concentration of the 4 compounds on the different samples (only present ratios) and there is no reference to the origin of theses samples on the Materials and methods section.
We gave identification of New Zealand specimen and we did add the following sentence:
The American Echium was propagated from the seeds sold by Outsidepride company, 915 N. Main St., Independence, OR 97351, in Bethel, Connecticut. Argentinian E. plantagineum was collected in B6600 Mercedes, Provincia de Buenos Aires, Argentina.
It was not our aim to determine concentration of the four compounds in the investigated samples.
Line 99- “peak showing an MS ion at m/z 398”. Please indicate the ionization mode. Does it correspond to a protonated molecule or refers to other mass adduct?
It was changed to:
MS ion at m/z 398 [M+H]+ corresponding to…
Line 105- The quality of figure 1 needs to be improved:
- Remove the colored background
- Remove the text box from the figure
- Add the units to the axis
- Improve the contents of the figure´s legend
It has been corrected, however we had to keep grey background for better contrast.
Line 115- The quality of figure 2 needs to be improved:
- Remove the colored background
- Remove the text and the arrows, leave only the letter attributed to each peak “A” or “B”
- Remove the outer lines
- Improve the contents of the figure´s legend
It has been corrected.
Line 120- “An NMR data analysis”, maybe replace by The NMR data analysis
OK, changed.
Line 129-158- The quality of the NMR description should be improved. The authors should always relate the carbon and proton chemical shifts to the atom number, also the name of the compounds should be presented as e.g. echimidine (1); echihumiline (2). Rephrase to achieve a better and less confusing description.
- “A detailed analysis of the 1H NMR spectrum of the constituent comprising peak A clearly indicates that two compounds, in the ratio of 3:1, are present.” How did you observed/calculated this?
The ratio was calculated from the integral values of the respective well-separated 1H NMR resonances. An appropriate clarification has been added to the manuscript.
- “In the minor component, the 1H signals at δH 6.75, 1.75 and 1.76, together with their adjacent carbons are characteristic for a tigloyl moiety, implying that this component is hydroxymyoscorpine (Figure 3)” Please assign the position to those proton chemical shifts and give a reference that corroborates your observations.
- “In the 1H NMR spectrum of the major component, the olefinic proton (H-18) signal is shielded to δH 5.57, as compared to H-19 of echimidine and hydroxymyoscorpine, whereas the methyl groups are both deshielded to δH 1.88 and 2.12. These chemical shifts are characteristic for a senecioic acid ester moiety, which permits the identification of the major component represented by peak A as echihumiline” – Improve to a clearer sentence, and provide a reference that corroborates your observations.
- “The 1H NMR spectrum of the second compound comprising peak B (…), as indicated by additional HMBC and COSY correlations”. Please assign the position of those chemical shifts and give a reference that corroborates your observations
- Make use of Table 1 throughout the text.
The manuscript has been corrected to include all the above points raised by the Referee.
Line 166-- The quality of figure 3 needs to be improved:
- Increase the pixel resolution of the structures
- Add the name of the compound below each structure e.g. echimidine (1); echihumiline (2)…
- Improve the contents of the figure´s legend
It has been corrected.
Line 180 – Improve the formatting of Table 1:
- Indicate the solvent, and frequency used for the NMR experiments
- Include the word “position” on the column of 1,2,3,…
- Present the carbon NMR data with only one decimal and introduce the number of attached protons using the C, CH, CH2, and CH3 notation.
- Use a space between the proton chemical shift data and the multiplicity of the signal
- Please indicate all the coupling constants
Table 1 has been corrected according to Referee’s suggestions.
Line 193 – Figure 4:
- Indicate the solvent, and frequency used for the NMR experiments
- Take care that the text does not overlap the peaks
Caption to Figure 4 has been amended according to Referee’s suggestion.
Lines 195-198- Is not clear what was the chromatographic relation between compounds 1-3 and compound 4. Please include experimental data and a reference that corroborates the assignment of compound 4 as echiumine. What is the importance of this finding?
We do not refer to the chromatographic relation between compounds 1-3 and compound 4, because compound 4 is well separated by CPC and HPLC from these three isomers. Neither we present experimental data as echiumine (compound 4) is well characterized in E. plantagineum. However the important thing is that in our biological assay compound 4 showed the highest hepatotoxicity.
Line 247- Replace UVIS by UV-VIS
It has been corrected.
Line 272- “These extracts were combined and quickly evaporated to dryness to avoid hydrolysis of ethyl acetate by the alkaloids.” Is this correct?
Yes, the statement is correct. Pyrrolizidine alkaloids are some of the strongest natural bases and on many occasions, we experienced it action on ethyl acetate. Ethyl acetate can be used for extraction, if care is taken to complete the extraction and subsequent evaporation of the solvent quickly. On numerous occasions, while performing Centrifugal Partition Chromatography, we observed that if the operation is finished within 3-4 hours, the alkaloids elute according to their partition co-efficients. If the CPC operation was interrupted for the night, the PAs were retained strongly in the aqueous phase due to pairing with acidic acid from hydrolysis of ethyl acetate.
Line 343 – “We believe that the two unreported alkaloids echihumiline and hydroxymyoscorpine have been overlooked in this species because MS does not differentiate isomers.” Could it be that these compounds have been overlooked because the standard experimental conditions are not appropriate like the liquid chromatography conditions used, or the lack of MS/MS data?
Yes, you are right. The same thoughts went through our minds, but we were somewhat shy to express these ideas as we highly regard many scientists who published on E. plantagineum and did not want to make overtly bold statements. However, we worked on the isolation of echimidine several times before and we also missed this issue. This discovery became possible because over time our technical capabilities improved and the number of observations resulting from repetitions led us to re-evaluation of the data. We will make suitable change to the manuscript.
Reviewer 3 Report
Review
Manuscript Title: Isolation of Echimidine and its C-7 isomers from Echium plantagineum L. and their hepatotoxic effect on rat hepatocytes.
[Page 9; Line 240]
Is it possible to add an explanation regarding the relationship between hepatotoxicity and extract composition? And also, the concerns regarding the finding on this research, is it concerning? Because despite the low composition of Echihumiline and Hydroxymyoscorpine, the IC50 is significantly lower compared to Echimidine.
[Page 9; Line 280]
Is the HPLC method for separating Echimidine and mixture of Echihumiline/Hydroxymyoscorpine validated? If it is validated it needs to be remarked as a beneficial addition to provide further in-depth food safety regulation. If it is not validated, it is good enough
[Page 2; Line 83]
Regarding the usage of the term “echimidine” as mixture of Echimidine, Echihumiline, and Hydroxymyoscorpine.
The usage of the term “echimidine” might confusing due to the similar words contained on the other constituents. Please consider to change the term which will be explained early on the manuscript. Maybe the term EM which stands for echimidine mixture will be useful.
The changed term will be also beneficial for explaining peaks on the caption of Figure 1, 2, and the respective hepatotoxicity on Figure 5. In addition, the term will also need to be explained on table footer on Table 2 to save space and facilitates reader better on digesting Table 2.
[Page 7; Line 191]
The secondary axis on 1H NMR spectra might be unnecessary
Author Response
Comments and Suggestions for Authors (3)
Manuscript Title: Isolation of Echimidine and its C-7 isomers from Echium plantagineum L. and their hepatotoxic effect on rat hepatocytes.
[Page 9; Line 240]
Is it possible to add an explanation regarding the relationship between hepatotoxicity and extract composition? And also, the concerns regarding the finding on this research, is it concerning? Because despite the low composition of Echihumiline and Hydroxymyoscorpine, the IC50 is significantly lower compared to Echimidine.
The values obtained for pure Echimidine and for its mixture with other isomers are statistically very similar. However, our experiments were conducted with rat hepatocytes and the results originated from one only experiment. For humans, the main source of echimidine is honey and contaminated herbal teas, which may be the source of chronic exposure. At this point, we cannot say with certainty that human hepatotocytes would produce results similar to those of rats. More study is warranted, considering that different batches of E. plantagineum may contain different rations of these isomers.
[Page 9; Line 280]
Is the HPLC method for separating Echimidine and mixture of Echihumiline/Hydroxymyoscorpine validated? If it is validated, it needs to be remarked as a beneficial addition to provide further in-depth food safety regulation. If it is not validated, it is good enough
The method was not validated because our work was strictly meant to support discovery and isolation, not quantification. However, during the work we tested several HPLC columns made by various producers that were designed to withstand high pH and nearly all were able to provide satisfactory or excellent resolution of the isomers of echimidine.
[Page 2; Line 83]
Regarding the usage of the term “echimidine” as mixture of Echimidine, Echihumiline, and Hydroxymyoscorpine.
The usage of the term “echimidine” might confusing due to the similar words contained on the other constituents. Please consider to change the term which will be explained early on the manuscript. Maybe the term EM which stands for echimidine mixture will be useful.
We agree that the naming used in the manuscript could be somewhat confusing. However we use the term “echimidine” peak where the word echimidine is in quatation mark. In this way we underline that the peak does not represents pure echimidine.
The changed term will be also beneficial for explaining peaks on the caption of Figure 1, 2, and the respective hepatotoxicity on Figure 5. In addition, the term will also need to be explained on table footer on Table 2 to save space and facilitates reader better on digesting Table 2.
As above.
[Page 7; Line 191]
The secondary axis on 1H NMR spectra might be unnecessary
A new figure is attached.